# β-Cyclodextrin Assisted Liquid–Liquid Microextraction Based on Solidification of the Floating Organic Droplets Method for Determination of Neonicotinoid Residues

**DOI:** 10.3390/molecules24213954

**Published:** 2019-10-31

**Authors:** Jitlada Vichapong, Khwankaew Moyakao, Rawikan Kachangoon, Rodjana Burakham, Yanawath Santaladchaiyakit, Supalax Srijaranai

**Affiliations:** 1Creative Chemistry and Innovation Research Unit, Department of Chemistry and Center of Excellence for Innovation in Chemistry, Faculty of Science, Mahasarakham University, Maha Sarakham 44150, Thailand; mai.2535@hotmail.com (K.M.); wittisit@gmail.com (R.K.); 2Materials Chemistry Research Center, Department of Chemistry and Center of Excellence for Innovation in Chemistry, Faculty of Science, Khon Kaen University, Khon Kaen 40002, Thailand; rodjbu@kku.ac.th (R.B.); supalax@kku.ac.th (S.S.); 3Department of Chemistry, Faculty of Engineering, Rajamangala University of Technology Isan, Khon Kaen Campus, Khon Kaen 40000, Thailand; sanyanawa@gmail.com

**Keywords:** β-cyclodextrin, liquid–liquid microextraction, HPLC, neonicotinoid insecticides

## Abstract

An efficient and environment-friendly microextraction method, namely, β-cyclodextrin assisted liquid–liquid microextraction, based on solidification of the floating organic droplets method coupled with HPLC is investigated for the sensitive determination of trace neonicotinoid pesticide residues. In this method, β-cyclodextrin is used as a disperser solvent, while 1-octanol is selected as an extraction solvent. β-cyclodextrins was found to decrease interfacial tension and increase the contact area between the organic and water phases with the help of centrifugation. A cloudy solution was rapidly formed and then centrifuged to complete phase separation. Various key parameters influencing extraction efficiency were systematically investigated and optimized; they include salt addition, concentration of β-cyclodextrin, and volume of extraction solvent (1-octanol). Under optimum conditions, good linearity was obtained with coefficient for determination (R^2^) greater than 0.99. A low limit of detection, high enrichment factor, and good recovery (83 – 132) were achieved. This proves that the proposed method can be applied to determine trace neonicotinoid pesticide residues in natural surface water samples.

## 1. Introduction

Sample preparation is an important and preliminary step in analytical processes. It helps provide both high selectivity and sensitivity for analysis by extracting, isolating, and preconcentrating trace amounts of the target analytes from complex sample matrices. Conventional sample preparation methods, such as liquid-liquid extraction (LLE) [1] and solid-phase extraction (SPE) [2] are the most extensively used for extraction of pesticides from various sample matrices. However, these techniques require large amounts of poisonous organic solvents, which are often hazardous. Moreover, the operation is time-consuming and tedious. To resolve this problem, research efforts has tried to demonstrate the development of environmentally friendly microextraction methods, including solid-phase microextraction (SPME) [3], stir bar sorptive extraction (SBSE) [4,5], ultrasound-assisted emulsification microextraction (USAEME) [6], salting-out assisted liquid–liquid extraction [7], and dispersive liquid–liquid microextraction (DLLME) [8,9] Generally, the DLLME process is based on a ternary component solvent system (a water-immiscible extractant, aqueous solution, and water-miscible disperser solvent), in which a cloudy solution is quickly formed after the rapid injection of the extraction and disperser solvents into the aqueous sample solution. The common extraction solvents used in this method are organic chlorinated solvents, such as chloroform and chlorobenzene. Because these halogenated solvents are not compatible with the mobile phase of the reversed-phase HPLC, it is evaporated to dryness before analysis by HPLC [10]. Thus, the dispersive liquid–liquid microextraction method based on solidification of floating organic drop (DLLME-SFO) [11] was introduced. In the DLLME-SFO method, 1-octanol, 1-dodecanol and toluene is chosen as a low-density extraction solvent, while organic solvent (methanol, acetonitrile) is also used. After centrifugation, the extraction solvent can be found floating on the top of the solution. DLLME-SFO is simple, easy to perate, is of low cost, high recovery, and utilizes low consumption of toxic organic solvents [12].

Recently, cyclodextrin was introduced into DLLME by Chen et al. [13] as a disperser solvent instead of an organic solvent. Cyclodextrin (CD) or cyclomaltoheptases are a well-known series of macro-cyclic oligosaccharides resulting from the degradation of starch by bacterial enzymes [14]. β-Cyclodextrin (β-CD) is an important cyclodextrin, composed of seven d-glucopyranose units, linked by α-1,4-glycosidic bonds [15]. It is a truncated cone-shaped macrocyclic molecule with a hydrophobic inner cavity (due to the presence of glycosidic oxygen bridges and hydrogen atoms) and a hydrophilic exterior (due to the presence of primary and secondary hydroxyl groups) [16,17], making β-CD an attractive host molecule in host-guest chemistry and supramolecular chemistry. It can stick selectively various organic, inorganic, and biological guest molecules as “a molecular shape sorter”, which are geometrically fit and less polar than water, into its cavity via non-covalent interactions to form stable host-guest inclusion complexes or nanostructured supramolecular assemblies. Thus, β-CD has the remarkable ability to recognize certain analytes in a highly selective and sensitive genre. β-Cyclodextrin (β-CD) has a remarkable capacity to select certain analytes because of its hollow truncated cone structure with a hydrophobic cavity and hydrophilic wall, allowing it to trap and hold targets of a certain size, with polarity in the cavity generating invertible and noncovalent inclusion complexes. Over recent years, the application on β-CD has gradually been extended and the host-guest type molecular recognition has been practically used in many fields, such as chemical separation [18], adsorbents [19], and food processing [20]. The first application by cyclodextrin-assisted dispersive liquid-liquid microextraction was reported in 2018 by Chen et al. [13] for preconcentration of carbamazim and clobazam. In this work, α-cyclodextrin was used as the dispersive solvent and chorinated solvent (chlorofrom) was selected as an extraction solvent. To be compatible with the mobile phase of HPLC, the extract was then evaporated to dryness and re-dissolved with a solvent before analysis.

Neonicotinoid insecticides, the principal alternatives to organophosphates and carbamates, are a class of broad-spectrum rapid-action insecticides used globally to control sucking insects [21,22]. Over the last few years, a resistance to existing insecticides has increased, in spite of the fact that neonicotinoids were presented as substances with several key attributes: high persistence, selective toxicity to arthropods, high water solubility and lower binding efficiencies to vertebrate compared to invertebrate receptors (low toxicity to humans and highly effective against insecticides) [23]. The widespread use of neonicotinoid insecticides at various stages of agricultural cultivation and during postharvest storage could give rise to serious health and safety risks [24]. They act as agonists at insect nicotinic acetylcholine receptors (nAChRs), which play an important role in synaptic transmission in the central nervous system [25]. Albeit the coming into force of market regulations seeking to limit pesticide usage in food products, in term of maximum residue limits (MRLs), such as the new European Union (EU) Regulation, a sensitive method for determining of neonicotinoid residues at low concentration levels is still needed to secure food quality and protect human health.

In this present work, we aim to present β-cyclodextrin based liquid–liquid microextraction based on solidification of floating organic droplets (β-cyclodextrin-LLME-SFO), followed by an analysis by HPLC with photodiode array detection for the preconcentration and simultaneous determination of neonicotinoid insecticide residues. Cyclodextrins are amphiphilic compounds with a hydrophilic shell and hydrophobic cavity and have been used as emulsifiers. They can decrease the surface tension between two phases by forming the organic solvent/cyclodextrin complexes at the liquid-liquid interface and enhancing the contact area between the organic and aqueous phase [10]. The parameters that affect the extraction performance of the microextraction method and HPLC performance are investigated and optimized. The applicability of the developed method for the determination of neonicotinoid insecticides in surface water samples is also demonstrated.

## 2. Results and Discussion

### 2.1. Optimization of the β-Cyclodextrin-LLME-SFO Procedure

Different parameters such as ionic strength, concentration of β-cyclodextrin, extraction solvent and its volume, and extraction time on the extraction efficiency of the analytes were optimized. In this experiment, these parameters were studied by one parameter at a time, while the other remaining factors were constant. The optimization was carried out in an aqueous solution (10.00 mL) containing 0.50 µg mL^-1^ of each analyte. Experiments to establish optimal conditions were repeated three times.

General, a suitable ionic strength decreases the solubility of the analytes in an aqueous sample solution while increasing their partitioning into the organic extraction phase. To study the effect of ionic strength on the proposed microextraction method, experiments were carried out with the addition of different electrolyte salts (NaCl, Na_2_SO_4_, Na_2_CO_3_ and CH_3_COONa) at 0.1 g and the results compared with that obtained from the process without salt addition. The experimental results are shown in Figure 1. It was found that the addition of Na_2_SO_4_ provided a higher extraction efficiency in term of peak area of the studied neonicotinoids, except acetamiprid and thiacloprid. It was found that with the use of various salts, such as NaCl, Na_2_SO_4_, CH_3_COONa, and without salt, the separation efficiency of the chromatogram response was not clear (data not shown). Therefore, Na_2_CO_3_ was used for further studies because it provides a high relative response in terms of peak area and good separation efficiency.

The amount of sodium carbonate for extraction efficiency of the studied neonicotinoids was also investigated and found to be within the range of 0.25–3.0 g. The relevant data are shown in Figure 2. It was found thatextraction efficiency, in terms of peak area of all target analytes, increased with a rise in sodium carbonate, up to 3.00 g. Beyond this point, it cannot dissolve in the solution, as it reaches its equilibrium. Therefore, 3.00 g of sodium carbonate was chosen for further studies.

Cyclodextrin was found to reduce the interfacial tension between two phases by forming organic solvent/cyclodextrin complexes as the liquid-liquid interface and increasing the contact area between two phases. β-CD is composed of glucopyranose units, which simultaneously occupy hydrophobic cavities and hydrophilic external surfaces, and has been widely used for separation [1]. For the above-mentioned procedure, β-CD was chosen as a disperser solvent. The effect of β-CD concentration on the extraction efficiency of the neonicotinoid standards was studied for the range of 3–45 mmol L^−1^. The relevant data are shown in Figure 3. There was an enhancement of extraction efficiency for all neonicotinoids when 15 mmol L^−1^ β-CD was added. This is because the droplet size of the 1-octanol/β-CD emulsion decreased with increasing β-CD concentration. Therefore, β-CD 15 mmol L^−1^ was selected as a disperser solvent.

The selection of extraction solvent and its volume has an important role in obtaining high recovery and enrichment factor in the DLLME-SFO system. The extraction solvent must meet several criteria: low volatility, low toxicity, low solubility in water, and solidification point near room temperature (in the range of 10–30 °C) in order to easily collect the solvent by solidification [26]. Based on these considerations, toluene, n-hexane, 1-dodecanol, and octanol were selected as potential extraction solvents for the study (data not shown). It was found that the chromatogram was unable to separate when toluene, *n*-hexane, and 1-dodecanol was added. Therefore, 1-octanol was selected because it has low density (0.8240 g mL^−1^) and its volumes were tested over the range of 50–300 μL. It was found that in 1-octanol volume less than 100 μL, the phase separation could not occur (Figure 4). Furthermore, by increasing the 1-octanol volume, peak area decreased owing to the dilution effect. Therefore, 1-octanol 100 μL was selected as optimum volume.

### 2.2. Analytical Performance of the Proposed Method

The proposed analytical method was evaluated under optimum conditions to extract the selected neonicotinoids by testing linearity, precision (RSD%), limits of detection (LODs), limits of quantification (LOQs), and enrichment factors (EFs), as shown in Table 1. The linearity of the proposed method was tested by preparing a series of spiked samples to establish the matrix-matched calibration curves. All the experiments were performed in triplicates. The calibration curve for each neonicotinoid was obtained by plotting the peak areas from their corresponding chromatograms versus the concentration of the target analyte. The method was linear in the range of 0.003–1 μg mL^−1^. The S/N = 3 was used for calculation of LOD and the S/N = 10 was used for calculation of LOQ. The LODs and LOQs were found to be in the range of 0.0001–0.0005 μg mL^−1^ and 0.0003–0.0015 μg mL^−1^, respectively. High precision was obtained with the RSDs of less than 10.99%. The efficiency of the developed method was evaluated in terms of EFs as the slope ratio of two calibration curves for analyte with and without the preconcentration (direct analysis). The EFs were in the range of 11–82. Figure 5 shows the chromatogram of standard neonicotinoids obtained by (a) without preconcentration: concentration of all standards was 0.50 µg mL^−1^, and (b) with preconcentration using β-cyclodextrin-LLME-SFO procedure: concentration of all standards was 0.50 µg mL^−1^.

### 2.3. Application to Real Samples

In order to evaluate the applicability of the developed method, the procedure was performed for the determination of target analytes in natural surface water samples. The results were shown in Table 2. The results indicated that there were no neonicotinoids in the studied samples. The samples were spiked with the target insecticides at different concentrations of 0.025, 0.050, and 0.100 µg mL^−1^, before extraction and analysis. The spiking recovery percentages of the target analytes in samples at different concentration levels are summarized in Table 2. It was found that the relative recoveries of the studied neonicotinoid insecticides were between 83% and 132% with RSD of less than 11.6% at the evaluated spiking concentration levels. Good recoveries were obtained, indicating that the developed method was effective and reliable for the analysis of the studied neonicotinoid insecticide residues in natural surface water sample matrix. Figure 6 shows a typical chromatogram of a natural surface water sample extracted by the proposed extraction method and analysis by HPLC.

### 2.4. Comparison of the Proposed Method with Other Methods

The performance of the proposed method was compared with other reported methods [27,28,29,30]. The results are shown in Table 3. This method provides high selectivity and sensitivity for the determination of analytes in complex matrices. Compared with other sample preparation methods, the proposed method offers several advantages, such as used alternative disperser solvent, low consumption of organic solvent, favorable reproducibility, and satisfactory relative recoveries. 

## 3. Experiments

### 3.1. Chemicals and Reagents

All chemicals were of analytical grade and deionized water (Millipore Waters, Milford, Massachusetts, USA) with resistivity of 18.2 MΩ.cm was used in the experiments. The analytical standards of neonicotinoid insecticides, including thiamethoxam, clothianidin, imidacloprid, and acetamiprid were obtained from Dr. Ehrenstorfer GmbH (Augsburg, Germany), and thiacloprid was purchased from Fluka (Leipzig, Germany). Common names and structures of the five neonicotinoids evaluated here are shown in Table 4. The stock solution of each insecticide (1000 μg mL^−1^) was prepared by dissolving it in methanol and storing at −20 °C until analysis. Working standard solutions were prepared by diluting the stock standard solution with water to obtain each concentration as required. β-Cyclodextrin (β-CD) was acquired from Fluka Chemie GmbH (Buchs, Switzerland). Methanol and acetonitrile of HPLC grade and 1-octanol were obtained from Merck (Darmstadt, Germany). Sodium chloride (NaCl), anhydrous sodium sulphate (anh. Na_2_SO_4_) and sodium carbonate (Na_2_CO_3_) were purchased from Ajax Finechem (North Shore, New Zealand); sodium acetate (CH_3_COONa) was obtained from Carlo Erba (Val de Reuil, France).

### 3.2. Apparatus and Chromatographic Conditions

The chromatographic separation was carried out on a Waters 1525 Binary LC system (Waters USA) equipped with Waters 2489 UV/Visible detector, a Rheodyne injector equipped with a sample loop of 20 µL. The Empower 3 software (Waters) was used for data acquisition. All separation was achieved on Chromolith^®^ HighResolution RP-18e (4.6 mm × 100 mm) with a flow rate of 0.5 mL min^-1^ at room temperature. The binary solvent system consisting of acetonitrile and water (26:74, *v/v*) was selected for separation of the target analytes. The injection volume was 20 µL and detection wavelength was set at 254 nm.

Five neonicotinoid insecticides were separated within 9 min with the elution order of thiamethoxam (*t_R_* = 4.54 min), clothianidin (*t_R_* = 5.30 min), imidacloprid (*t_R_* = 5.76 min), acetamiprid (*t_R_* = 6.45 min), and thiacloprid (*t_R_* = 8.91 min).

### 3.3. β-Cyclodextrin-LLME-SFO Procedure

Figure 7 shows a schematic diagram of the proposed microextraction method. First, 3.0 g of Na_2_CO_3_ was added into a 10-mL screw cap test tube containing 10 mL of sample or standard solution. Then, β-CD was added to the tube. The mixture was centrifuged at 1500 rpm for 7 min. After that, the extraction solvent (1-octanol) was rapidly injected into the tube before vortexing for 30 s. It was then centrifuged at 1500 rpm for 10 min to complete the phase separation and the reconstituted solution was observed at the bottom of the solution. The target analytes in aqueous sample were extracted as fine droplets that settled on the top of the solution. Then, 20 µL of the phase was kept and directly injected into the HPLC.

### 3.4. Sample Preparation

Natural surface water samples were taken from different areas located near rice fields in Maha Sarakham province, northeastern Thailand, filtered using Whatman filter paper No. 42, and then passed through 0.45 µm nylon membrane filter before extraction using the β-Cyclodextrin LLME-SFO procedure.

## 4. Conclusions

β-cyclodextrin assisted liquid–liquid microextraction based on solidification of the floating organic droplets (β-cyclodextrin-LLME-SFO) method coupled with HPLC was successfully applied for the efficient enrichment of trace neonicotinoids from natural surface water samples. The method is simple, quick, effective, and offers convenient operation. The developed method has good analytical features, providing a low limit of detection in the range of 0.10–0.50 µg L^−1^ for all compounds, which is below the acceptable MRLs for neonicotinoids. High preconcentration factor, good recovery, and high reproducibility were also obtained. This strategy has the capability to be adapted for the preconcentration of other trace organic pollutants from different samples.

## Figures and Tables

**Figure 1 molecules-24-03954-f001:**
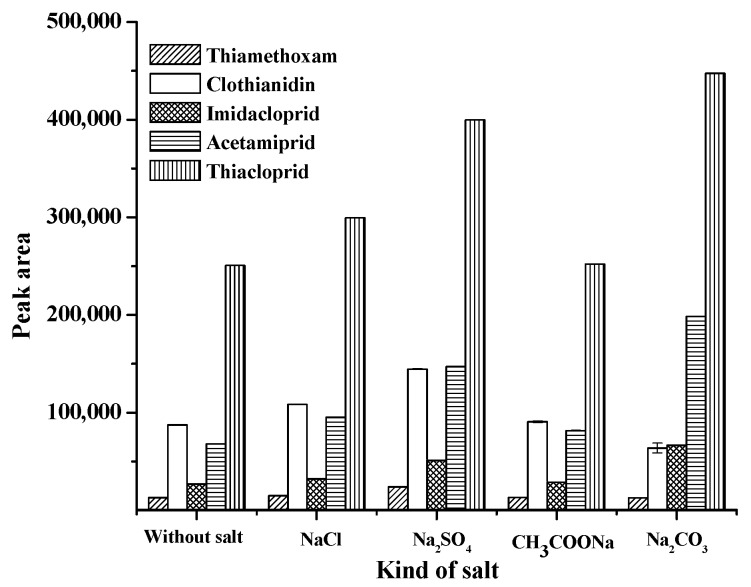
Effect of salt addition on the extraction of studied neonicotinoids.

**Figure 2 molecules-24-03954-f002:**
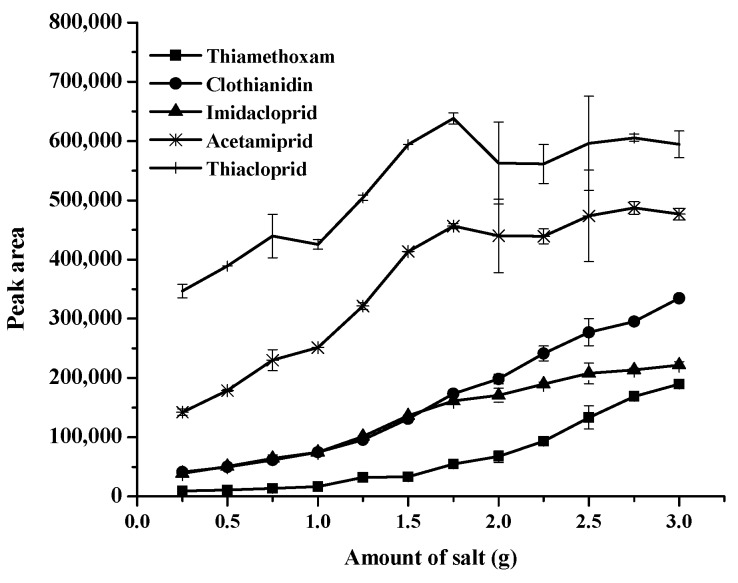
Effect of the amount of salt on the extraction of studied neonicotinoids.

**Figure 3 molecules-24-03954-f003:**
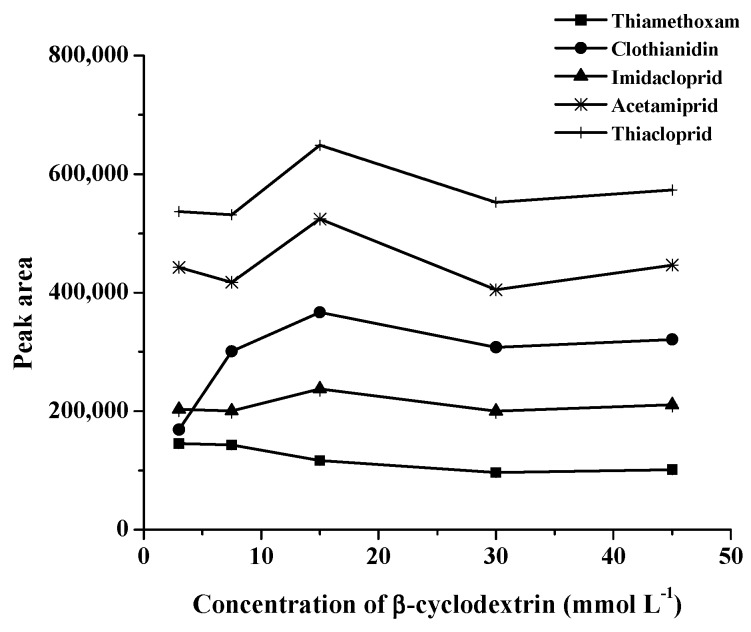
Effect of concentration of β-cyclodextrin on the extraction of studied neonicotinoids.

**Figure 4 molecules-24-03954-f004:**
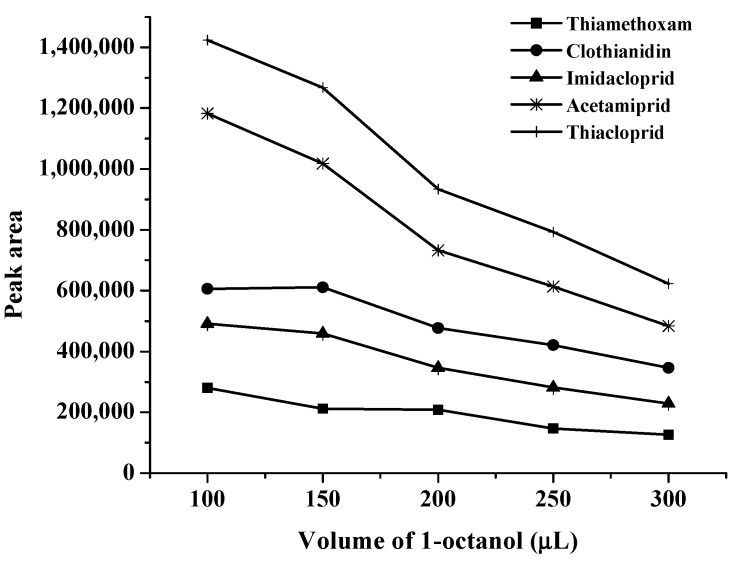
Effect of volume of 1-octanol on the extraction of studied neonicotinoids.

**Figure 5 molecules-24-03954-f005:**
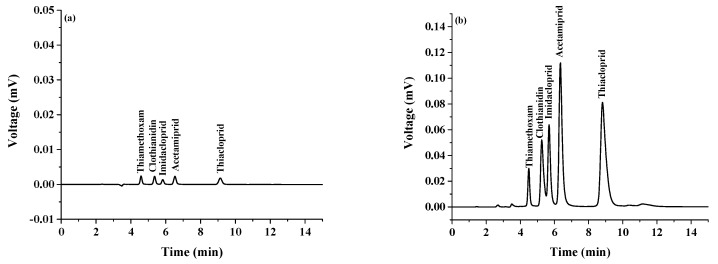
Chromatogram of standard neonicotinoids obtained by (**a**) without preconcentration and (**b**) with β-cyclodextrin-LLME-SFO procedure: concentration of all standards was 0.50 µg mL^−1^.

**Figure 6 molecules-24-03954-f006:**
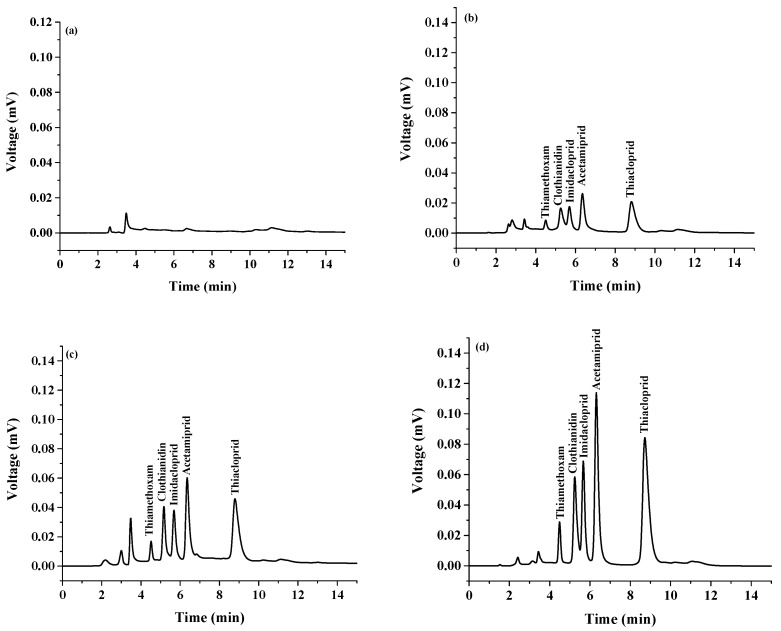
Typical chromatograms of (**a**) water sample, (**b**) water sample spiked at 0.025 µg mL^−1^ of each insecticide, (**c**) water sample spiked at 0.050 µg mL^−1^ of each insecticide, and (**d**) water sample spiked at 0.100 µg mL^−1^ of each insecticide, extracted by the proposed extraction method and analysis by HPLC.

**Figure 7 molecules-24-03954-f007:**
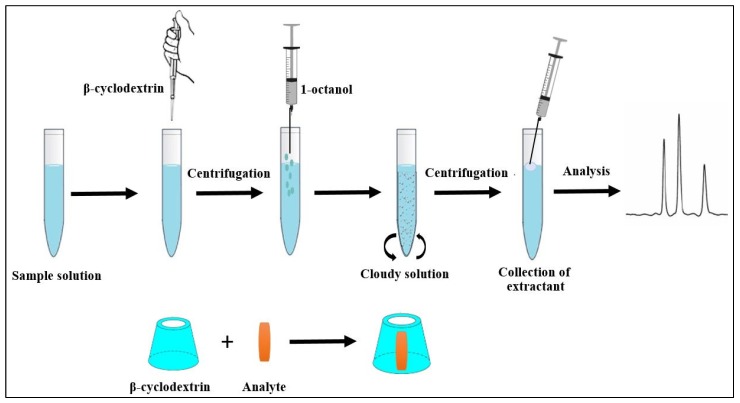
Schematic diagram of the proposed microextraction method.

**Table 1 molecules-24-03954-t001:** Analytical performance of the β-Cyclodextrin-LLME-SFO method.

Pesticide	β-Cyclodextrin-LLME-SFO							
	Linear Range (μg mL^−1^)	LOD (μg mL^−1^)	LOQ (μ g mL^−1^)	Intra-day (%RSD, n = 5)		Inter-day (%RSD, n = 3 × 5)		EF
				t_R_	Peak Area	t_R_	Peak Area	
Thiamethoxam	0.0015–1	0.0005	0.0015	0.71	7.15	0.91	10.99	10.69
Clothianidin	0.0006–1	0.0002	0.0006	0.68	3.68	0.86	6.84	25.93
Imidacloprid	0.0003–1	0.0002	0.0003	0.65	7.82	0.81	9.43	52.53
Acetamiprid	0.0003–1	0.0001	0.0003	0.65	8.38	0.84	9.50	44.69
Thiacloprid	0.0003–1	0.0001	0.0003	0.75	9.34	0.79	9.83	81.62

**Table 2 molecules-24-03954-t002:** Recovery obtained for the determination of neonicotinoid insecticides in natural surface water samples (n = 3).

Sample	Spiked (µg mL^−1^)	% Recoveries at Different Spiked Levels (% RSD)				
		Thiamethoxam	Clothianidin	Imidacloprid	Acetamiprid	Thiacloprid
Surface water I	0.000	-	-	-	-	-
	0.025	76.65 (3.48)	100.33 (1.15)	75.09 (1.06)	93.10 (1.06)	73.57 (0.15)
	0.050	87.18 (2.67)	120.88 (11.6)	97.26 (8.68)	124.82 (2.61)	83.42 (0.80)
	0.100	99.69 (1.95)	114.96 (0.97)	91.49 (0.58)	132.42 (1.36)	86.88 (2.68)
Surface water II	0.000	-	-	-	-	-
	0.025	81.86 (1.05)	92.82 (1.07)	100.36 (2.94)	128.76 (0.95)	95.18 (2.08)
	0.050	84.86 (2.45)	103.74 (0.63)	96.61 (3.30)	127.79 (4.61)	83.40 (4.70)
	0.100	92.03 (2.87)	109.38 (4.97)	99.14 (3.15)	128.48 (4.53)	90.73 (5.86)
Surface water III	0.000	-	-	-	-	-
	0.025	80.92 (0.17)	99.77 (0.17)	94.79 (0.81)	126.45 (3.57)	87.99 (1.20)
	0.050	87.80 (0.53)	104.48 (1.86)	98.08 (0.80)	120.39 (0.83)	87.41 (0.83)
	0.100	95.19 (3.69)	106.68 (0.73)	99.01 (1.77)	124.33 (4.59)	84.13 (4.06)

**Table 3 molecules-24-03954-t003:** Comparison of the proposed method and other methods to determine neonicotinoids.

Method	Sample	LOD	Linearity	Recovery (%)	Ref.
**VSLLME-SFO**	Fruit juice and water	0.1–0.5 (µg L^−1^)	0.0005–5 (µg mL^−1^)	85–105	[28]
**SPE**	Drinking water	0.01 µg L^−1^	0–1 (mg L^−1^)	95–104	[27]
**DSPE**	Water samples	0.02–0.4 (ng mL^−1^)	10–500 (ng mL^−1^)	7–119.0	[29]
**VA-D-µ-SPE**	Fruit juice and natural surface water	0.005–0.065 ng mL^−1^	0.5–1000 ng mL^−1^	70–138	[30]
**β-cyclodextrin-LLME-SFO**	Natural surface water	0.10–0.50 (µg L^−1^)	0.003–1.00 (mg L^−1^)	83–132	This study

VSLLME-SFO: Vortex-assisted surfactant-enhanced-emulsification liquid–liquid microextraction with solidification of floating organic droplet. SPE: Solid Phase Extraction. DSPE: Dispersive Solid Phase Extraction. VA-D-µ-SPE: Vortex-assisted Dispersive Micro Solid-Phase Extraction. β-cyclodextrin-LLME-SFO: β-cyclodextrin assisted liquid–liquid microextraction based on solidification of the floating organic droplets method.

**Table 4 molecules-24-03954-t004:** Properties of the studied neonicotinoid insecticides with regard to other chemical classes.

Neonicotinoid Insecticide	Water Solubility (mg L^−1^) at 20 °C	Log K_OW_	Structure
Thiamethoxam	4100	−0.13	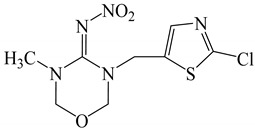
Imidacloprid	610	0.57	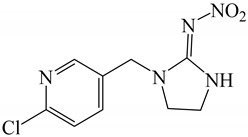
Clothianidin	340	0.91	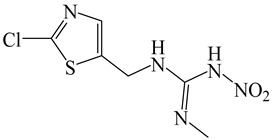
Acetamiprid	2950	0.80	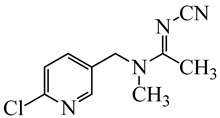
Thiacloprid	184	1.26	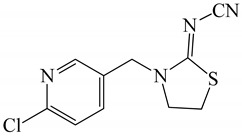

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
