# Peer review of "β-Cyclodextrin Assisted Liquid–Liquid Microextraction Based on Solidification of the Floating Organic Droplets Method for Determination of Neonicotinoid Residues"

_molecules, 2019, doi:10.3390/molecules24213954_

Round 1
Reviewer 1 Report
I review the manuscript, β-cyclodextrin assisted liquid–liquid microextraction based on solidification of floating organic droplets method for determination of neonicotinoid residues (molecules-623812). The manuscript is an interesting idea but some concepts must be reviewed. The abstract and introduction sections must be reviewed for a native speaker, the main ideas are not clear.
Some critical points:
1) Please use the template of molecules.
2) The Tittle mentions that the technique is a solidification of floating organic droplets. However, the solvent employed (1-octanol) is liquid at room temperature. In the text the authors mention that beta-cyclodextrin was used as surfactant, why the methodology was not considered as a dispersive liquid-liquid microextraction methodology. Additionally, the 1- octanol was never solidified; it must be only a floating organic drop.
3) Please include the limit of detection in the abstract.
4) Please include some information about the analytes: base structure, pKa values, log P. This is relevant to understand the extraction and separation processes.
5) Please include information about inclusion complexes between the analytes and beta-cyclodextrin.
6) It is important mention that beta-cyclodextrin is solid, the authors mention that it is used as solvent but in order to added it into the extraction system it must be dissolved. It was used as an additive not as a solvent. Please review the concepts of the extraction process.
7) Please include the extraction-elution conditions used during the optimization.
8) Please include a discussion about the pH value employed during extraction. The behavior corresponds to analytes with basic functional groups. This point in congruent with point 4.
9) Line 149. Beta-cyclodextrin is solid not a solvent.
10) The mobile phase for HPLC was ACN-water (26:74), did the authors evaluate the solubility of 1-octanol in this solvent? In my own experience a higher volume of CAN was required in order to obtain a homogenous solution.
11) Validation section must be reviewed carefully. Precision (inter and intra-day) must be evaluated at three different levels. LOD values are not congruent (HPLC and LLME) with traditional methodologies. 5 micrograms/L in not achieved in HPLC-UV without any sample pre-concentration technique.
12) In order to guarantee applicability of the methodology in surface waters, a matrix effect assay must be performed. The analytical sensitivity (deionized water and surface water) must be statically compared in order to quantify in the conditions proposed.
13) Please include similar analytical matrices and Table 3, it is not correct to compare other matrices.
14) Section 3.2 (line 247). Please include the particle size if the stationary phase.
15) Section 3.3. Please include the optimal conditions found for the proposed methodology.
Author Response
Dear Editor,
We thank you and the reviewers very much for the valuable comments. The manuscript has been carefully revised accordingly to the comments as in the following: Manuscript entitled “β-cyclodextrin assisted liquid–liquid microextraction based on solidification of floating organic droplets method for determination of neonicotinoid residues”
All changes are highlighted in the revised manuscript using red colored.
Thank you very much for consideration of our manuscript.
Yours sincerely,
Authors

Reviewer 2 Report
In this manuscript, a β-cyclodextrin assisted liquid-liquid microextraction based on solidification of floating organic droplets method have been presented and used with HPLC for sensitive determination of trace neonicotinoid pesticide residues in water samples. This method is environmentally friendly and interesting. The review suggestions are listed in the following.
In this work, β-cyclodextrin was adopted as disperser solvent. Why not using other macro-cyclic oligosaccharides, including cyclomaltoheptases and other type cyclodextrins, or some derivatives of β-cyclodextrin? Some fatty acids, 1-dodecanol and 1- undecanol are also widely used as extraction solvent in DLLME-SFO methods. Why the authors selected 1-octanol? More detailed reasons are needed. Fig. 4 & 5: Baseline separation of clothianidin and imidacloprid has not been achieved. The chromatographic conditions for this separation should be further optimized. The data for optimization of 1-octanol volume should be provided as supplementary information. Table 1: The unit “ng mL-1” is more suitable than “μg mL-1” for linear range, LOD and LOQ. The full name of the method in Table 3 should be added in the footnote. Using the abbreviation without full name would make the reader confused. There are some grammatical errors or spelling mistakes in the manuscript. This manuscript needs polishing. For example, Line 83: carbamazim should be carbamazepine. Line 84: chorinated should be chlorinated.Author Response
Dear Editor,
We thank you and the reviewers very much for the valuable comments. The manuscript has been carefully revised accordingly to the comments as in the following: Manuscript entitled “β-cyclodextrin assisted liquid–liquid microextraction based on solidification of floating organic droplets method for determination of neonicotinoid residues”
All changes are highlighted in the revised manuscript using red colored.
Thank you very much for consideration of our manuscript.
Yours sincerely,
Authors

Reviewer 3 Report
Dear Authors,
one comment for you, maybe in keywords you adding "surface water sample" and that all. thank you
Author Response

(The authors gave the same response as above.)

Round 2
Reviewer 1 Report
I review the manuscript “β-cyclodextrin assisted liquid–liquid microextraction based on solidification of floating organic droplets method for determination of neonicotinoid residues” (Molecules-623812). The edition topics were considered but the analytical points were not considered. The manuscript requires additional experiments in order to conclude the applicability of the method.
Some critical points
The author did not include the structure, pKa and log de P values of the analytes. It was did not included the justification of the extraction process. Please justify the lack of an internal standard.
The validation section must include a precision study (in spiked samples) or at least discuss the results in Table 2. Some data are not congruent, it is normal to observe a higher RSD values at lower concentrations. Additionally, it must be included an intra- inter-day study.
The linear range is considered from LOQ.
The matrix effect was not evaluated appropriated. It is required to analyse and compare properly the analytical sensitivity of standard solution prepared in deionized water and spiked blank samples.
The Table 3 was not useful to compare the methodology proposed because the samples are not similar.
Author Response
Response to Reviewer 1 Comments
Point 1: I review the manuscript “β-cyclodextrin assisted liquid–liquid microextraction based on solidification of floating organic droplets method for determination of neonicotinoid residues” (Molecules-623812). The edition topics were considered but the analytical points were not considered. The manuscript requires additional experiments in order to conclude the applicability of the method.
Some critical points
The author did not include the structure, pKa and log de P values of the analytes. It was did not included the justification of the extraction process.
Response 1: Thank you very much for your kind suggestion. Change was made as suggested.
Point 2: Please justify the lack of an internal standard.
Response 2: Thank you very much for your kind suggestion. The method of internal standards is used to improve the precision of quantitative analysis. In this method, high precision was obtained therefore we didn’t used an internal standard.
Point 3: The validation section must include a precision study (in spiked samples) or at least discuss the results in Table 2. Some data are not congruent, it is normal to observe a higher RSD values at lower concentrations. Additionally, it must be included an intra- inter-day study.
The linear range is considered from LOQ.
Response 3: Thank you very much. Change was made as suggested.
Point 4: The matrix effect was not evaluated appropriated. It is required to analyse and compare properly the analytical sensitivity of standard solution prepared in deionized water and spiked blank samples.
Response 4: Thank you very much for your kind suggestion. The recovery experiments were carried out to investigate the method accuracy and precision. The samples were spiked with standard neonicotinoids before analysis by the whole analytical process as proposed. It was found that, the relative recoveries of the studied neonicotinoid insecticides were between 83 and 132% with the RSDs of less than 11.6%, at the evaluated spiking concentration levels. Good recoveries were obtained, indicating that the developed method was effective and reliable for the analysis of the studied neonicotinoid insecticide residues in natural surface water sample matrix. This is chromatogram of standard in water.
And this is chromatogram of standard in surface water sample.
Point 5: The Table 3 was not useful to compare the methodology proposed because the samples are not similar.
Response 5: Thank you very much. Change was made as suggested.

Reviewer 2 Report
This manuscript has not been well revised. The response to reviewers’ comment is perfunctory. Additional experiments are still required to improve the proposed method.
β-cyclodextrin was adopted as disperser solvent. But the authors should point out the reason why not using other macro-cyclic oligosaccharides, including cyclomaltoheptases and other type cyclodextrins, or some derivatives of β-cyclodextrin. Line 139-140: the data about the investigation of different kinds of extraction solvent must be shown in this manuscript. Line 142: As the previous comment, the data regarding the optimization of 1-octanol volume also need to be shown in this manuscript. Point 6 in the previous comment: “The full name of the method in Table 3 should be added in the footnote. Using the abbreviation without full name would make the reader confused.” The revision has not been made in the revised manuscript. Mostimportant ofall, baseline separation of clothianidin and imidacloprid has not been achieved. The chromatographic conditions for this separation should be further optimized. Therefore, additional experiments are still required to improve the proposed method.Author Response
Response to Reviewer 2 Comments
Point 1: This manuscript has not been well revised. The response to reviewers’ comment is perfunctory. Additional experiments are still required to improve the proposed method.
β-cyclodextrin was adopted as disperser solvent. But the authors should point out the reason why not using other macro-cyclic oligosaccharides, including cyclomaltoheptases and other type cyclodextrins, or some derivatives of β-cyclodextrin.
Response 1: Thank you very much for your kind suggestion. Detail was shown in revised manuscript (Line 81-83).
Point 2: Line 139-140: the data about the investigation of different kinds of extraction solvent must be shown in this manuscript.
Response 2: Thank you very much. Change was made as suggested.
Point 3: Line 142: As the previous comment, the data regarding the optimization of 1-octanol volume also need to be shown in this manuscript.
Response 3: Thank you very much. Change was made as suggested.
Point 4: Point 6 in the previous comment: “The full name of the method in Table 3 should be added in the footnote. Using the abbreviation without full name would make the reader confused.” The revision has not been made in the revised manuscript.
Response 4: Thank you very much. Change was made as suggested.
Point 5: Most important of all, baseline separation of clothianidin and imidacloprid has not been achieved. The chromatographic conditions for this separation should be further optimized. Therefore, additional experiments are still required to improve the proposed method.
Response 5: Thank you very much for your kind suggestion.
In further optimized, we will change the mobile phase composition from 25% acetonitrile in water to 35% acetonitrile in water to separate clothianidin and imidacloprid
